# Knee Joint Preservation in Tactical Athletes: A Comprehensive Approach Based upon Lesion Location and Restoration of the Osteochondral Unit

**DOI:** 10.3390/bioengineering11030246

**Published:** 2024-03-01

**Authors:** Daniel J. Cognetti, Mikalyn T. Defoor, Tony T. Yuan, Andrew J. Sheean

**Affiliations:** 1Department of Orthopedic Surgery, Brooke Army Medical Center, 3551 Roger Brooke Drive, San Antonio, TX 78234, USA; mtdefoor@gmail.com (M.T.D.); ajsheean@gmail.com (A.J.S.); 2Advanced Exposures Diagnostics, Interventions and Biosecurity Group, 59 Medical Wing, Lackland Air Force Base, San Antonio, TX 78236, USA; tony.yuan@usuhs.edu; 3Center for Biotechnology (4D Bio3), Uniformed Services University of the Health Sciences, Bethesda, MD 20814, USA

**Keywords:** cartilage restoration, marrow stimulation, matrix associated chondrocyte implantation, osteochondral allograft, osteochondral autograft, osteoarthritis

## Abstract

The unique physical demands of tactical athletes put immense stress on the knee joint, making these individuals susceptible to injury. In order to ensure operational readiness, management options must restore and preserve the native architecture and minimize downtime, while optimizing functionality. Osteochondral lesions (OCL) of the knee have long been acknowledged as significant sources of knee pain and functional deficits. The management of OCL is predicated on certain injury characteristics, including lesion location and the extent of subchondral disease. Techniques such as marrow stimulation, allograft and autologous chondrocyte implantation are examined in detail, with a focus on their application and suitability in tactical athlete populations. Moreover, the restoration of the osteochondral unit (OCU) is highlighted as a central aspect of knee joint preservation. The discussion encompasses the biomechanical considerations and outcomes associated with various cartilage restoration techniques. Factors influencing procedure selection, including lesion size, location, and patient-specific variables, are thoroughly examined. Additionally, the review underscores the critical role of post-operative rehabilitation and conditioning programs in optimizing outcomes. Strengthening the surrounding musculature, enhancing joint stability, and refining movement patterns are paramount in facilitating the successful integration of preservation procedures. This narrative review aims to provide a comprehensive resource for surgeons, engineers, and sports medicine practitioners engaged in the care of tactical athletes and the field of cartilage restoration. The integration of advanced preservation techniques and tailored rehabilitation protocols offers a promising avenue for sustaining knee joint health and function in this demanding population.

## 1. Introduction

Military service exacts a physical toll that uniquely challenges the musculoskeletal integrity of individuals in uniform. Articular cartilage plays a pivotal role in facilitating mobility for service members navigating the multifaceted demands of their roles. However, these same demands also put the articular cartilage at risk of injury, with chondral injuries of the knee being particularly prevalent among tactical athletes, including an increased risk of high-grade and multifocal articular cartilage defects as compared to the general population [1]. These injuries present a conspicuous threat to the individual service member as well as the military’s broader operational readiness. As such, it is imperative to refine approaches to knee joint preservation that not only mitigate immediate impairments but also ensure the long-term functional capacity of affected service members.

While various conservative treatments are often advocated initially, their success is often limited in individuals with larger, more extensive lesions, as well as those attempting to return to more demanding physical activities. Although advancements in the field of orthopedics and tissue engineering have provided a myriad of techniques and technologies to address chondral injuries, variability in post-operative outcomes limits the strength of existing treatment algorithms. However, despite the variable clinical presentations of chondral injuries and results of interventions, a certain measure of consensus related to cartilage restoration has emerged. Central to this evolving consensus is our understanding of the osteochondral unit, a composite structure comprised of layers of articular cartilage and the underlying subchondral bone. This unit represents a fundamental biomechanical entity within the joint, and its preservation or restoration based on the extent of injury is paramount for functional recovery. In light of these considerations, this narrative review aims to present a comprehensive approach to knee joint preservation in the tactical athlete. By delineating treatment strategies based on lesion location and emphasizing the restoration of the osteochondral unit, we endeavor to provide a framework that optimizes outcomes and prolongs servicemembers’ careers, while highlighting some emerging treatment modalities.

## 2. Components of the Osteochondral Unit

Restoration of the osteochondral unit (OCU) is a central concept in knee joint preservation. The OCU comprises the articular cartilage, the calcified cartilage, and the subchondral bone. Injury to any of these components can disrupt the delicate balance required for smooth joint motion and load-bearing capacity.

Articular cartilage is absent of blood vessels, lymphatics and nerves. It is composed mostly of extracellular matrix and water, with chondrocytes dispersed throughout. The extracellular matrix is made up of collagen, proteoglycans and glycoproteins, which allow for the biomechanical properties of cartilage [2,3]. More specifically, aggrecan, the largest and most abundant proteoglycan in cartilage, attracts water which allows cartilage to facilitate load transmission to subchondral bone, while collagen, the most abundant structural macromolecule, provides tensile strength and mechanical integrity. Chondrocytes establish, maintain and repair the ECM structure of articular cartilage. However, they have limited ability to replicate and limited capacity for healing on a macroscopic level. This is in part related to their sparseness, making up only 2% of the total volume of cartilage, and in part due to mostly anaerobic metabolism given their limited supply of oxygen/nutrition, which occurs via diffusion.

## 3. Layers of the Osteochondral Unit

Articular cartilage can be broken down into three distinct layers. The thin surface layer is referred to as the superficial layer. This layer constitutes approximately 20% of the cartilage thickness and is notable for flattened chondrocytes and high collagen content, aligned parallel to the weight bearing surface [4]. This collagen arrangement helps create cartilages smooth gliding surface and gives the superficial layer its increased tensile strength, which helps protect the inner layers from sheer stress. The next layer, referred to as the middle or transitional zone, represents approximately 50% of the cartilage thickness. This layer contains a relatively lower density of chondrocytes, but the chondrocytes here have a higher relative concentration of organelles to aid in extracellular matrix synthesis. Additionally, the collagen fiber arrangement is randomly oriented in this zone, helping to provide some resistance to compressive force. Finally, the deepest layer of the articular cartilage is referred to as the deep zone. This layer is approximately 30% of the articular cartilage thickness and is most important for dissipating compressive forces. Within the deep layer, the collagen fibers are oriented perpendicularly to the joint surface to fulfill this role. Directly underneath this layer, the tidemark differentiates the articular cartilage from the calcified cartilage, with calcified cartilage serving as the strong attachment site between the articular cartilage and subchondral bone (Figure 1) [5]. Aside from serving as the bedrock of articular cartilage, the subchondral bone appears to serve an important role in cartilage restoration (and osteoarthritis progression), with subchondral edema being predictive of worse outcomes and higher failure rates of surface based restoration procedures [6,7,8,9,10].

## 4. Classification of Articular Cartilage Injuries

In order to appropriately treat articular cartilage injuries, we must be able to reliably classify their size and severity, including the extent of subchondral bone involvement. While the size of the lesion can be estimated on MRI, its full extent is often better appreciated at the time of arthroscopy. However, aside from size, quantitative assessments regarding the degree of injury can also be made with MRI and arthroscopy. There are several different schemes with which to classify these lesions. The Outerbridge classification and International Cartilage Repair Society (ICRS) are the most common. Both are graded from 0–4. In both, Grade 0 refers to normal cartilage, whereas Grade 4 refers to full thickness cartilage loss with exposed subchondral bone. Within the Outerbridge classification, Grade I refers to cartilage softening and swelling, Grade II refers to partial thickness defects less than 50% of the thickness of the articular cartilage with a diameter less than 1.5 cm, and Grade III refers to deep fissuring greater than 50% thickness, without exposed subchondral bone.

## 5. Aging and Articular Cartilage Degeneration

Aging results in predictable changes to articular cartilage and subchondral bone [10,11]. This includes increasing stiffness from non-enzymatic glycosylation of the extracellular matrix, decreasing water content and chondrocyte number. Importantly, although these changes are felt to predispose to cartilage degeneration, they are distinct from those seen with cartilage degeneration and osteoarthritis. In osteoarthritis, initial cartilage fibrillation and proteoglycan degradation result in increased cartilage permeability. This then leads to increased water content and, in turn, decreased cartilage stiffness. Reductions in the organization of collagen meshwork similarly contribute to changes in the mechanical properties of cartilage [12]. Although chondrocytes may attempt to initiate a reparative response, with increased matrix synthesis in response to degeneration, loss of chondrocyte number and downregulation of their anabolic response eventually results in progressive cartilage loss and the clinical diagnosis of osteoarthritis [13].

## 6. Preoperative Evaluation and Imaging

In evaluating patients with known or suspected articular cartilage injuries, a standard set of imaging and evaluations should be performed. Any history of knee or patellar instability should be elucidated and assessed with examination. Coronal plane limb alignment should be assessed both clinically and as measured with hip to ankle radiographs, as abnormalities in the weight bearing axis should be considered in the development of a comprehensive management approach. Generalized ligamentous laxity should also be assessed and quantified using Beighton criteria.

Advanced imaging typically includes non-contrast magnetic resonance imaging (MRI) of the knee, wherein the extent and depth of the articular cartilage injury can best be determined. Although gadolinium enhanced MRI has been advocated for identifying early articular cartilage changes with enhanced sensitivity, it is not routinely utilized in the authors’ practices [14]. On MRI, subchondral bone involvement can be assessed via T2 or short tau inversion recovery (STIR) sequences. Bone edema on these sequences would steer the treatment algorithm away from surface-based cartilage procedures and toward osteochondral autograft or allograft treatments. In instances of patellar instability, radiographic parameters, such as tibial tubercle to trochlear groove distance, patellar height and trochlear dysplasia, should be assessed, as should the status of the cruciate and collateral ligament ligaments and menisci in tibiofemoral chondral injuries. Occasionally, computerized tomography (CT) scans may also be ordered to further assess osseous involvement. This is particularly relevant in younger tactical athletes who have acute or chronic osteochondritis dissecans lesions, those unable to obtain MRI, or in revision settings where there is questionable integrity of the subchondral bone.

## 7. Lesion Location-Specific Considerations

### 7.1. Patellofemoral Joint Cartilage Lesion

Articular cartilage injuries in the patellofemoral joint can be frustrating for patients and surgeons alike. They can cause debilitating anterior knee pain and often patients present with a component of their pain related to patellar tendinopathy, which may mask the underlying structural issue. Due to this, these patients often present to orthopedic surgeons after a protracted course of disability, despite the relatively high sensitivity and specificity of MRI for detecting these lesions [15]. While conservative treatment is still advocated as first line management, many tactical athletes with patellofemoral lesions seeking to return to high level activities eventually require surgical management.

The patellar cartilage has a lower compressive aggregate modulus, greater thickness (4–6 mm peak thickness in males [16]) and higher fluid permeability than trochlear femoral cartilage. These differences may predispose the patella to earlier degenerative changes, whereas the femoral trochlea cartilage exhibits the highest stiffness of any region in the knee [17,18]. These adaptations are likely to arise from the distinct mechanical environment of the patellofemoral compartment, which is characterized by a higher degree of shear as compared to the rest of the knee.

As a result, historically there have been concerns about the suitably of certain cartilage restoration procedures in the patellofemoral joint and the possibility of higher rates of failures due to sheer. However, a systematic review of 59 studies analyzing a variety of cartilage restoration procedures showed improved clinical outcomes postoperatively with low rates of complications [19]. Nevertheless, some reservations remain regarding the use of osteochondral allografts (OCA) and particularly dual OCAs for bipolar lesions, where there is articular cartilage damage on both the patella and trochlea [20,21,22]. Yet recent systematic reviews have demonstrated positive clinical outcomes with patellofemoral OCAs with high rates of return to sport [23,24], with specific indications being large defects, abnormal subchondral bone, and/or revisions.

### 7.2. Tibiofemoral Joint Cartilage Lesion

Femoral condyle lesions are typically readily accessible arthroscopically or through a medial or lateral arthrotomy. They can be managed with the full complement of described techniques in the surgical management section below, but one variable to consider that is not discussed elsewhere is the relative containment of the lesion. If a lesion is present within the central aspect of the condyle and circumferentially surrounded by walls of cartilage, it is said to be contained, whereas a lesion along the femoral notch or outer aspect of the condyle that does not have intact walls is said to be uncontained. Uncontained lesions, particularly those with less than 50% containment, are less suitable for marrow stimulation or injectable technologies because the biologic material will not congeal and remain in place as readily. However, even when utilizing osteochondral plugs, lesions with less than 50% containment can still be difficult to treat. In these situations, surgeons should be prepared to augment press-fit fixation of plugs with compression screws if there is any concern about plug security [25].

Although femoral-based lesions have received substantially more attention, tibial-based lesions are quite common and often occur in concert with femoral chondral injuries. Similar to lesions in the patellofemoral joint, bipolar lesions in the tibiofemoral compartment typically have more modest outcomes than isolated lesions, with relatively high failure rates [26]. However, concomitant treatment of tibial lesions with marrow stimulation following osteochondral allograft transplantation (OCA) of the femur has not been shown to effect outcomes, leaving uncertainty as to the utility of addressing tibial lesions in these instances [27]. From a practical perspective, posteriorly based tibial lesions can at times be difficult to access and treat with the full complement of restoration techniques, and thus retrograde drilling can be a useful alternative form of marrow stimulation.

Lastly, surgeons must consider the contribution of limb alignment to chondral injury and its impact on the success of a restoration procedure in the medial and lateral tibiofemoral compartment [28]. Along the same line, a prior meniscal injury or debridement can also predispose to chondral injury, while putting patients at risk of failure of a cartilage restoration procedure [29]. Meniscal deficiency should be promptly addressed with meniscal repair, if a repairable tear is present, or possibly with a meniscal allograft transplantation (MAT), but it is clear that, if a MAT is indicated, the chances of a tactical athlete returning to full duty are exceedingly low [30].

## 8. Non-Surgical Management

In treating tactical athletes with articular cartilage injuries there are a number of social and occupational factors that may steer treatment, such as upcoming deployments, duty station changes, occupational demands and retirement or separation dates.

The decision to pursue non-operative management depends on these social factors as well as the previously attempted non-operative modalities, in addition to the degree of pathology. It is important to note that the degree of articular pathology presents along a spectrum from acute and/or isolated articular cartilage lesions to chronic, multifocal lesions. The patients at each end of this spectrum are treated differently. This review focuses mostly on the former with attempts at knee joint preservation, whereas the treatment for the latter group, more consistent with diffuse degeneration and osteoarthritis, is mostly palliative, given a lack of efficacious long-term solutions short of a knee arthroplasty.

In the vast majority of cases, service members should first undergo a dedicated course of physical therapy/rehabilitation, activity modification and/or rest. Additionally, service members should be counseled on modifiable risk factors, such as weight loss, tobacco use and androgenic steroid use [31,32]. In one study of obese patients who lost weight over 48 months, there was significantly lower progression of cartilage degeneration on MRI as compared to those with stable body weight [33]. Similarly, based on gadolinium enhanced MRIs, weight loss also appears to preferentially improve the quality and quantity of articular cartilage in the medial compartment [34]. Finally, a study utilizing a lower-body positive pressure treadmill (AlterG, Fremont, CA, USA) which modulates bodyweight, showed that, with walking, the applied body weight influences the degree of articular cartilage catabolism [35]. This lends further credibility to the utility of weight loss as a treatment strategy, while helping to support the use of off-loading treadmills as standard of care for rehabilitation in articular cartilage injuries.

In conjunction with therapy, other non-operative treatments, such as compressive wraps, off-loader braces, local modalities, transcutaneous electrical stimulation and injections, may be trialed. Although their long-term efficacy is likely to be limited, injections can be a useful treatment adjunct to ameliorate patient’s symptoms, while also potentially aiding in diagnosis. However, despite them often being advocated prior to surgery, there is no evidence that they can reverse or fully repair existing chondral damage. At present, injections are most easily classified into three groups: visco-supplementation (hyaluronic acid), corticosteroids and biologics (platelet rich plasma, bone marrow aspirate, adipose derived, etc.).

There are several available visco-supplementation formulations and sometimes they are collectively referred to as cartilage analogues or synthetic cartilage. They are most commonly utilized in mild to moderate osteoarthritis, but despite several proposed mechanisms of action, their in vivo effect and efficacy remain uncertain [36], with the 2021 Evidence-Based Clinical Practice Guidelines from the American Academy of Orthopedic Surgery stating that hyaluronic acid injections are not recommended for routine use in the treatment of symptomatic osteoarthritis of the knee [37].

Corticosteroids constitute the longest standing class of injections. As potent anti-inflammatories, they can reliably alleviate some degree of pain [38] and may even allow tactical athletes to complete training cycles, missions or deployments. However, their effect is of limited duration and regular or recurrent use should be undertaken cautiously [39].

Biologic injections, such as platelet rich plasma (PRP), adipose derived and synovial derived preparations, represent a growing area of interest. However, due to uncertain efficacy within specific applications, they are sometimes associated with substantial out of pocket expenses. PRP injections, for example, may cost several hundred dollars in the civilian market, but are covered by Tricare within the Military Health System and are readily accessible to tactical athletes. PRP is derived from a patient’s own blood. The drawn blood is centrifuged, separating it into distinct, concentrated layers. Specific layers containing supra-physiologic levels of platelets and growth factors are then syphoned off and injected back into the patient. While initially touted for its potential reparative effects and natural derivation from one’s own body, there is no evidence that in isolation it can restore cartilage defects and its clinical efficacy is the subject of substantial debate [40]. While one systematic review from Dold et al. in 2014 [41] concluded that there “is a paucity of data supporting the use of PRP for the management of focal traumatic osteochondral defects”, much of the preceding literature and more recent literature investigating an osteoarthritis population has also not been able to show a benefit of its use [42]. Similarly, preclinical and clinical studies regarding the other biologics, such as autologous lipoaspirate and synovial derived injections, have shown some positive early effects, but high level evidence is lacking and most of these studies are again focused on an osteoarthritis population, rather than those with localized osteochondral defects [43].

Finally, it is important to touch on the topic of stem cells as many tactical athletes, and others too, will come in asking for such treatments to cure their cartilage injury. We counsel these patients that at present there are no clinically available injection treatments which can recreate their native cartilage and, although at times various biologic treatments are referred to as stem cell treatments, the currently available treatments (PRP, bone marrow aspirate, adipose preparations) contain such low concentrations of stem cells that the practice of referring to them in this manner at all has been discouraged. This point notwithstanding, there is great interest in harnessing the potential of stem cells [44]. This includes utilizing them as cell based therapies or as similarly derived exosomes with signaling factors to halt the progression of degeneration and even potentially reverse it [45,46,47,48,49,50].

## 9. Surgical Management

The surgical management of chondral lesions is generally predicated upon size, lesion location within the knee and the extent of involvement within the osteochondral unit. Figure 2 depicts the authors’ preferred algorithmic approach to the surgical management of knee chondral lesions.

OATs: Osteochondral autograft transplantation, MS (+): Marrow stimulation (enhanced marrow stimulation), OCA: Osteochondral allograft transplantation, AMIC: Autologous matrix induced chondrogenesis, MACT: Matrix-associated autologous chondrocyte transplantation, cm: Centimeter.

### 9.1. Chondroplasty

The simplest form of surgical treatment is an arthroscopic chondroplasty, which involves the removal of loose edges and/or flaps of articular cartilage. A chondroplasty can be utilized as definitive management for lower grade and small lesions (≤1 cm) with a relatively fast rehabilitation period and no weight bearing restrictions. This fact makes chondroplasty particularly preferable among tactical athletes as it accommodates a rapid return to unrestricted activity. Putting it in the context of another high performing athletic population, Scillia et al. showed that 67% of NFL athletes were able to return to play at 8 months postoperatively [51]. Aside from serving as an expedited and potentially definitive management option, chondroplasty is also commonly utilized during staged diagnostic arthroscopies, wherein the size of the lesion is further evaluated for later definitive management.

Historically, chondroplasty has been the most common cartilage-based procedure performed, accounting for 80% of cartilage restoration procedures. However there are growing concerns about its durability [52,53]. Additionally, the populations undergoing chondroplasty, as demonstrated by the currently available evidence, are noticeably older (47 years of age for chondroplasty versus 41 years for marrow stimulation and 32 years for osteochondral grafting), revealing that chondroplasty is a palliative treatment option, but that younger patients with sizeable and symptomatic lesions are likely to be best served with a higher order operation [53,54].

### 9.2. Marrow Stimulation

Marrow stimulation can be performed either through an arthroscopic or open technique. The cartilage defect is debrided to remove the calcified cartilage layer and the borders of the defect are sharply defined in order to demarcate lesion borders from healthy surrounding cartilage. Small holes into the subchondral bone are then either punched or drilled to a depth of 3–4 mm to stimulate elution of marrow elements into the base of the created defect. Removal of the calcified cartilage layer appears to allow for optimal healing [55], while the perforations create channels for bone marrow cells to migrate into the defect and form fibrocartilage. While arthroscopic marrow stimulation alone represents a low cost and minimally invasive option, variable clinical outcomes have been observed in the literature, especially in larger lesions or in high-demand athletes. In a systematic review by Goyal et al. of Level I and II evidence, marrow stimulation provided good clinical outcomes in the short-term for small lesions with lower postoperative demands, but after 5 years there were high rates of treatment failure regardless of lesion size [56]. Similarly, in a randomized control trial by Gudas et al. comparing marrow stimulation and osteochondral autograft transplantation (OATs), only 52% of athletes returned to sport after marrow stimulation, compared to 93% in the OATs group [57]. On a national level, concerns about the modest outcomes and durability of marrow stimulation [58] are reflected in the decreasing number of marrow-stimulating procedures being performed by recently trained orthopedic surgeons sitting for the American Board of Orthopedic Surgery Part II examination [52]. For these reasons, marrow stimulation in isolation should be utilized cautiously in the tactical athlete owing to concerns regarding the long-term durability of the procedure and its effects on the condition of the underlying subchondral bone.

### 9.3. Juvenile Allograft Cartilage

Juvenile Allograft Cartilage implantation utilizes particulated cadaveric juvenile allograft cartilage to replace damaged tissue. Implantation can be done open or with arthroscopic visualization without utilization of fluid. After demarcating the borders of the lesion, the calcified cartilage layer is removed. The particulate cartilage is placed in the defect and covered with fibrin glue. From a technical perspective, a single packet of particulated cartilage can cover a 2.5 cm^2^ defect, with a recommended fill ratio of particulate cartilage of 50% in order to avoid overgrowth of the new cartilage.

The logistics of this technique are favorable, as it can be performed in a single-stage fashion. Other advantages include the use of juvenile cartilage (donors 13 years of age or less) which demonstrates 100-fold more chondrocyte activity than adult cartilage [59]. Additionally, histologic analyses of eight patients at two years follow-up revealed a mixture of hyaline and fibrocartilage, with a predominance of type II collagen, more closely replicating native articular cartilage [60]. However, although several short-term clinical studies [60,61,62] have revealed overall good outcomes, including within the patellofemoral joint, long-term outcome data is lacking. Additionally, although it represents an “off-the-shelf option”, one of the biggest drawbacks is that the shelf life is limited to less than a month, given that it is fresh tissue, in comparison to other frozen or decellularized extracellular allograft options, which have up to a 5-year shelf life.

### 9.4. Cartilage Extracellular Matrix Allograft

Similar to particulate juvenile cartilage, allograft cartilage extracellular matrix (ECM) or other protein scaffolds can be utilized alongside marrow stimulation, platelet rich plasma (PRP) or bone marrow concentrate (BMAC) to fill chondral defects [63]. The acellular nature substantially extends the shelf-life, but it is important to note that the grafts themselves lack viable chondrocytes, making them strictly chondro-conductive and inductive [64].

There are a number of commercially available scaffolds and they are best characterized according to their structure, either as particulate ECMs, hydrogels or sheets. Their composition varies based on the manufacturer, but they are mostly a mixture of collagen, collagen oligomeric protein (COMP) and other ECM products.

The implantation of these scaffolds is sometimes collectively referred to as enhanced marrow stimulation (marrow stimulation +) or as autologous matrix-induced chondrogenesis (AMIC) because they utilize ECM and an autologous source of cells, such as marrow stimulation or BMAC, to induce chondrogenesis. However, these terms can at times be non-specific and/or confused with autologous chondrocyte implantation (ACI), matrix-associated autologous chondrocyte transplantation (MACT), or their proprietary contingent, matrix-induced/-associated autologous chondrocyte implantation (MACI). In any regard, the use of cartilage ECM technologies has been shown to augment traditional marrow stimulation with improved histological scoring compared to marrow stimulation alone in an equine model [65]. Short-term clinical outcomes have also been good overall, but outcome studies within the knee remain relatively limited [66,67]. However, one systematic review comparing AMIC to ACI at 40 month follow-up, found that AMIC had better Lysholm and IKDC scores by a margin exceeding the minimum clinically important difference, with noticeably lower complication rates than ACI [68]. This may in part be related to AMIC procedures, being a single stage operation, as compared to ACI, which requires two stages; however, directed comparative studies of these two techniques are needed prior to making conclusions about the superiority of one technique over the other.

### 9.5. Matrix-Associated Autologous Chondrocyte Transplantation (MACT)

These procedures entail the harvest of a patient’s own cartilage from a healthy, non-weight-bearing area, followed by in vitro expansion on a scaffold and eventual re-implantation into the lesion site. This method offers the potential for hyaline cartilage regeneration, as opposed to fibrocartilage, but it does require a staged approach. Additionally, in the authors’ practices, significant preoperative subchondral edema serves as a relative contraindication for this technique due to concerns over compromised outcomes and higher failure rates, with the authors’ favoring an osteochondral plug technique in these situations [8].

The most commonly utilized and well-studied MACT technique is MACI (Matrix-Associated Chondrocyte Implantation), which is a proprietary acronym that is often utilized more generally to refer to this larger subpopulation of procedures. There have been several iterations and advancements of this technology over time. The first generation MACI required a periosteal patch to be sewn in place over the defect and the chondrocytes were applied underneath, the second generation technique utilized a cellularized matrix patch, but it still had to be sutured into place, and now the third-generation patch is secured with fibrin glue [69]. At 5 years follow up, 98% of patients were satisfied with their level of knee pain with 93% satisfaction at 10 years [70,71]. This included more than 80% of patients having good to excellent cartilage infill at 5 and 10 years [70,71,72].

More recently, an increasing appreciation for the complex three-dimensional structure of articular cartilage has led to new technologies. These include Novocart 3D, a bilayer collagen scaffold, and Novocart Inject, a hydrogel, on which to grow autologous cartilage for later implantation. Several two-year clinical outcome and imaging studies have shown positive results, including one multicenter study looking at large defects ranging from 4–12 cm^2^ [73,74,75,76]. However, just like the juvenile allograft cartilage, there is some refinement needed to avoid overgrowth [74].

### 9.6. Osteochondral Autograft Transfer (OAT)

In this procedure, a plug of healthy cartilage and underlying bone from the patient is taken from a non-weight-bearing part of the joint and transferred into the damaged area. This technique restores the OCU architecture and, given that it is autograft, there is a decreased risk of immunogenic rejection and improved restoration of normal hyaline cartilage. Other advantages include that it is a single stage operation with substantially lower cost than other techniques, given that it obviates the need for the purchase of a scaffold or allograft.

In terms of outcomes, a randomized control trial comparing OATs to marrow stimulation demonstrated excellent clinical outcomes and return to sport in patients undergoing OATs as compared to marrow stimulation at a mean follow-up of 37 months [57]. Additionally, a systematic review of studies on OATs demonstrated successful outcomes in 72% of patients at long-term follow-up [77]. However, OAT is limited in its application by the size of the defect, given the limited availability of area for harvest and associated donor site morbidity. Specific to a tactical athlete population, donor site-related pain is often a persistent complaint.

Typically, lesions up to 2 cm in diameter are considered suitable for OATs. In order to fill larger defects, mosaicplasty is also an option, wherein several smaller osteochondral plugs can be harvested and implanted to cover the defect. While there is some concern about higher failure rates in these operations, a randomized control trial in pediatric patients with osteochondritis dissecans lesions showed superiority of mosaic-type OATs over marrow stimulation [78].

### 9.7. Osteochondral Allograft Transplantation (OCA)

Osteochondral allograft transplantation is similar to osteochondral autograft transfer but involves using harvested plugs from a cadaveric donor. This technique has several nuances with regard to obtaining grafts, graft storage and preparation, which are important to the success of the operation.

Fresh allograft tissue suitable for OCA is harvested within 24 h of the death of a donor and must be size matched to the recipient via CT scan. Typically, donors are recommended to be between 15 and 40 years of age and must have otherwise healthy articular cartilage in the area of interest. These grafts are commercially available from tissue banks, but given that they are fresh, non-frozen and require size matching, obtaining a graft can take several weeks or months and, after a suitable graft is identified, then the operation must occur in a time-sensitive manner to ensure graft viability at the time of implantation. The uncertainty associated with this wait time is a significant drawback for high-tempo tactical athletes and their commands, although this operation is most commonly utilized in larger defects and revision scenarios, where the previous detailed modalities are less suitable.

Proper graft processing and storage are essential, as both can affect graft viability, shelf life and immunogenicity. Gross et al. [79] demonstrated that success of the operation depends on the presence of viable chondrocytes in the superficial layer, the incorporation of the bony plug and intact extracellular matrix. It is recommended that fresh tissue be implanted within 28 days of harvest and ideally within two weeks, in order to maximize the number of viable chondrocytes. Grafts are typically stored in media anywhere between 4 °C and 37 °C. However, hypothermic conditions (4 °C) may decrease chondrocyte viability and biomechanical properties, whereas grafts stored at physiologic temperatures have demonstrated maintained viability. In practice, the Missouri Osteochondral Allograft Preservation System (MOPS), which applies several of the above best practices in a comprehensive protocol, has been shown to extend OCA shelf life and chondrocyte viability, as well improving short-term clinical outcomes [20,80,81]. For similar reasons, cryopreserved and fresh-frozen allografts are less ideal candidates for OCA implantation [82].

Osseous integration of OCA plugs has also recently begun to receive more attention, given its implications on the success of the operation [83]. Several studies have investigated pretreatment protocols for OCA grafts, such as saline lavage and utilization of BMAC to help with bone integration [84]. While one study, based on radiographs, showed BMAC improved integration [85], two other studies utilizing MRI did not show any advantage [86,87]. However, there is substantial heterogeneity within and between the studies, which limits the ability to draw meaningful conclusions about the applied techniques. Similarly, various graft washing techniques have been advocated, including pulse lavage, to decrease graft immunogenicity and bone marrow contents, but thus far the results have been conflicting, without a clear demonstration of clinical benefit [84,88,89,90].

There have been a number of clinical studies on OCA outcomes both at short-term and long-term follow-up. A systematic review by Familiari et al. of 19 studies demonstrated a mean survival rates of 86.7% and 78.7% at five and ten years, respectively [83]. Similarly, another systematic review investigating return to sport rates from 13 studies, demonstrated that most athletes (75–82%) were able to return after OCA with improvements exceeding minimal clinically important differences in sport-specific outcomes [91]. However, a study of large OCA grafts (mean defect size: 4.87 cm^2^) in a military population demonstrated a disappointing rate of return to physical activity, with only 42% returning to military activity [92].

Mosaicplasty involves the transplantation of multiple small osteochondral grafts to the damaged area. This technique may allow a more anatomically accurate replacement of both the articular cartilage and subchondral bone and/or may allow surgeons to span large defects. However, for autograft mosaicplasty the availability of suitable donor sites can limit its application and for allograft mosaicplasty versus single plug OCA there is concern that the additional interfaces may lead to higher complication rates [93].

Finally, synthetic osteochondral plugs are a subject of interest and are currently under further development, representing a welcome solution to the constraints of limited tissue availability associated with allograft donors and the limited area amenable for autograft harvest. In a rabbit osteochondral defect model, polyvinyl alcohol (PVA) hydrogels were shown to be biocompatible, with maintenance of the implant properties in vivo for at least 12 weeks [94]. Human studies have similarly demonstrated good biocompatibility and even shown promotion of hyaline cartilage regeneration [95,96]. Clinical outcome studies with PVA hydrogels have also demonstrated promising early results, with one study of 18 patients demonstrating improved knee outcomes up to 6 months [97]. However, the outcomes do appear to diminish at 12 months, although another study of 18 patients at 5–8 year follow up demonstrated acceptable outcomes when the implants were utilized for what were subjectively considered appropriate indications [98].

An important and illuminating component of the preclinical analyses of PVA hydrogels has been biomechanical in nature. Although Sismondo et al. initially demonstrated restoration of contact pressures in the knee with hydrogels [99], the loads utilized were below that of physiologic loading, and Chen et al. [100] later demonstrated that PVA hydrogels in isolation did not improve contact mechanics, as compared to femoral osteochondral defects without treatment. However, with the addition of a porous titanium base underneath the PVA implant, contact mechanics were improved. This has since dovetailed the current technologies into a new direction of interest, including bipolar implants, which more closely replicate both native bone and cartilage and, more recently, tri-layer implants have also been tested, which further recreate the native OCU [101].

As therapeutic cellular and scaffold-based solutions have continued to develop, there has been a growing array of techniques and technologies which combine both domains to optimize cartilage restoration. Excitement surrounds these modalities and particularly three dimensional, multi-layer scaffolds, which can signal both structurally and chemically [101].

## 10. Rehabilitation Guidelines

In the context of tactical athletes, selecting the most appropriate technique for OCU restoration requires careful consideration of lesion size, location, and individual patient factors, but post-operative rehabilitation and physical conditioning are equally vital components of the comprehensive approach. These programs focus on strengthening the surrounding musculature, improving joint stability, and optimizing functional movement patterns to ensure the long-term success of the procedure.

In the initial phase (0 to 6 weeks) following cartilage restoration procedures, the primary objective is to safeguard the graft. Weight-bearing allowances depend on the specific lesion location, with a universal aim to minimize shear or compressive stress on the transplanted area.

Generally, for interventions in the patellofemoral compartment, surgeons may allow immediate weight bearing in full extension, while more conservative weight bearing precautions are placed on lesions within the medial or lateral compartment. However, for those with patellar or trochlear lesions, some recommend restricted knee flexion (<45–90°) for the initial 4 to 6 weeks. Individuals undergoing treatment for femoral condyle or tibial plateau lesions typically adhere to toe-touch weight-bearing for 4–6 weeks. Although there is no consensus on post-operative bracing, within our military population we routinely advocate for and apply hinged knee braces for cartilage restoration procedures. If a brace is employed for these cases, it is gradually adjusted in 20° increments as quadriceps control improves. Weight-bearing and range of motion (ROM) limitations are tailored to the specific procedures conducted for concurrent pathology (e.g., osteotomy, ligament reconstruction, meniscal transplant, or repair). Notably, weight-bearing after ACI has been shown to be safe as early as 6 weeks postoperatively [102].

The second phase of rehabilitation (6 to 12 weeks) is geared towards restoring the patient’s capacity to engage in everyday functional activities. Typically, braces are discontinued once the patient demonstrates satisfactory quadriceps control and can execute a straight leg raise without extension lag. Although some surgeons suggest use of an unloader brace for up to 4 months postoperatively to alleviate stress on the affected compartment, especially applicable for patients with bipolar lesions, we do not routinely ascribe to this practice. Patients progress towards achieving full ROM, normalized gait, and the initiation of closed-chain exercises, along with gentle strengthening.

The final phase of rehabilitation (beyond 3 months) is contingent on the patient’s specific goals and expectations. For individuals with more modest goals, aiming to resume daily activities without pain after a salvage procedure, a maintenance home exercise program can facilitate a gradual return to these activities. More typically, in the case of tactical athletes seeking to return to duty, this phase emphasizes advanced strengthening, core stabilization, proprioception, and a gradual reintegration into duty-specific training. Individuals should be advised against engaging in activities that impose excessive impact loading on the new cartilage tissue, particularly within the first year post-surgery. Ideally, high-loading activities should be postponed until 6 to 12 months postoperatively. Athletes must demonstrate full ROM, ligamentous stability, absence of effusion, and outstanding dynamic strength prior to a thorough consideration of the risks and benefits associated with returning to the highest level of activity. The decision to return should adhere to a rigorous set of criteria, to be determined by the treating surgeon and physiotherapists, such as testing or functional testing (single leg hop, squat, patient psychological preparedness, etc.).

## 11. Conclusions

The treatment of knee chondral injuries among tactical athletes necessitates a comprehensive approach based upon the location and extent of the lesion. A number of promising biologic and synthetic options are available to surgeons based upon the condition of the osteochondral unit. The unique physical demands of tactical athletes should compel ongoing innovation in the field and future research should focus on enhancing the reliability and durability of clinical outcomes of cartilage restoration surgery.

## Figures and Tables

**Figure 1 bioengineering-11-00246-f001:**
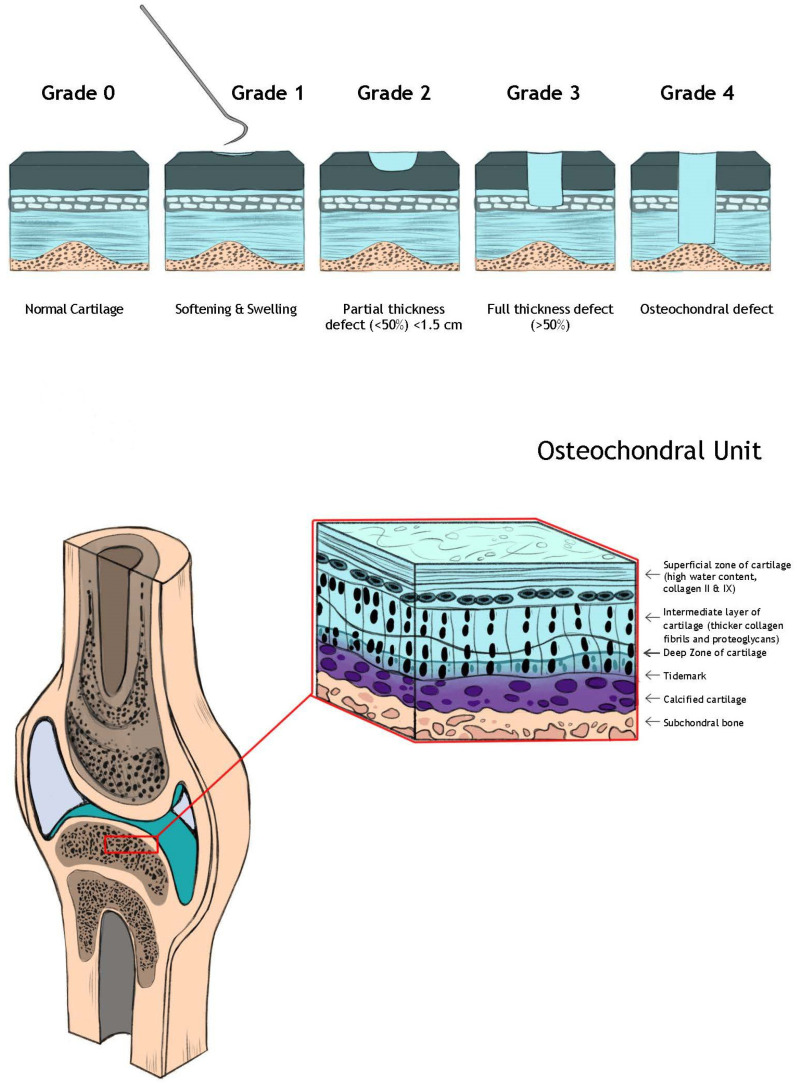
Structure of the Osteochondral Unit and Injury Classification.

**Figure 2 bioengineering-11-00246-f002:**
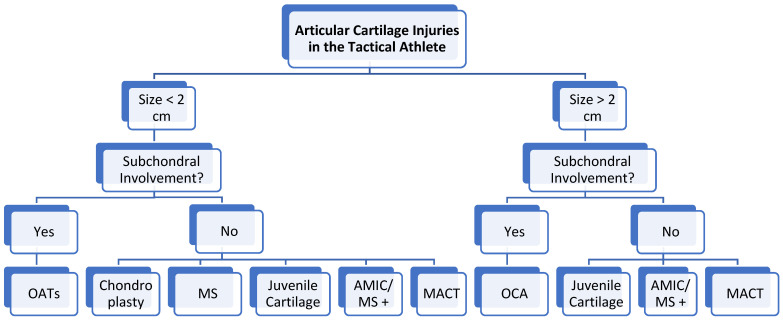
Treatment Algorithm for Articular Cartilage Injuries in the Tactical Athlete.

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
