# Peer review of "Knee Joint Preservation in Tactical Athletes: A Comprehensive Approach Based upon Lesion Location and Restoration of the Osteochondral Unit"

_bioengineering, 2024, doi:10.3390/bioengineering11030246_

Round 1

Reviewer 1 Report

Comments and Suggestions for Authors

This “review” provides a narrative on osteochondral repair, focusing on tactical athletes. First, please specify the type of review (i.e., “narrative review”) asap in the abstract as this is indicative of a different order of quality as compared to e.g. a “systemic review”. Osteochondral repair has recently been reviewed abundantly and excellently - even systemically. I am afraid that I do not directly see any novel insights this paper would contribute. To this end, in my eyes, the authors fail a bit short in pointing out what differentiates tactical athletes from other athletes or age-matched sportive individuals. From my point of view, the release of 2-3 “new” articles since the most recently cited review do not justify another review on the same topic. Additionally, in general, all other relevant articles should be referenced, like e.g. Christensen et al. 2021, 10.1177/1947603520904757; Husen et al. 2022, 10.1016/j.jcjp.2022.100049 – a systemic review; Lesage et al. 2022, 10.1002/advs.202200050; Chen et al. 2023, 10.1016/j.bioactmat.2023.04.016; Nakagawa et al. 10.1038/s41536-023-00335-x, etc etc

Importantly, the paper would benefit from a message (i.e. clear conclusion), while earlier review articles already concluded that “the methods are promising, but large randomized controlled studies are needed”, the present “aims” of this narrative review are vague.

Author Response

Knee Joint Preservation in Tactical Athletes: A Comprehensive Approach Based Upon Lesion Location and Restoration of the Osteochondral Unit

bioengineering-2800617

Reviewer 1:

This “review” provides a narrative on osteochondral repair, focusing on tactical athletes. First, please specify the type of review (i.e., “narrative review”) asap in the abstract as this is indicative of a different order of quality as compared to e.g. a “systemic review”. Osteochondral repair has recently been reviewed abundantly and excellently - even systemically. I am afraid that I do not directly see any novel insights this paper would contribute. To this end, in my eyes, the authors fail a bit short in pointing out what differentiates tactical athletes from other athletes or age-matched sportive individuals. From my point of view, the release of 2-3 “new” articles since the most recently cited review do not justify another review on the same topic. Additionally, in general, all other relevant articles should be referenced, like e.g. Christensen et al. 2021, 10.1177/1947603520904757; Husen et al. 2022, 10.1016/j.jcjp.2022.100049 – a systemic review; Lesage et al. 2022, 10.1002/advs.202200050; Chen et al. 2023, 10.1016/j.bioactmat.2023.04.016; Nakagawa et al. 10.1038/s41536-023-00335-x, etc etc

Narrative review, added

Importantly, the paper would benefit from a message (i.e. clear conclusion), while earlier review articles already concluded that “the methods are promising, but large randomized controlled studies are needed”, the present “aims” of this narrative review are vague.

The authors have added substantial detail to the article. We thank the reviewer for the proposed citations, we have read through them and added in details or ideas from each as appropriate. This includes a section which further details the unique treatment considerations in tactical athletes. We feel that our current conclusions collectively summarize the important aspects of emphasis within the review.

Reviewer 2 Report

Comments and Suggestions for Authors

The work proposed by the authors is a comprehensive narrative synthesis of the concept of "osteochondral unit," with particular reference to the knee, its pathophysiology, and current treatment indications in case of damage to this unit. While theoretically focused on the so-called "tactical athletes," a closer examination of the manuscript reveals a lack of specific details essential for honing the discussed themes toward this distinct population. In other words, the concepts presented remain broad and lack the necessary refinement tailored explicitly to this category of athletes.

Moreover, the study is entirely oriented towards surgical treatment, overlooking the current understanding that knee osteochondral damage, particularly in case of chronic or overuse injuries, is often more effectively managed through non-surgical approaches including physical therapies, rehabilitation, pharmacological interventions, and infiltrative treatments (i.e.: hyaluronic acid, PRP, mesenchymal stem cell, etc.).

I suggest a meticulous revision of the manuscript, with a sharper focus on specific physical carachteristics and performance-related issues specific to tactical athletes. Furthermore, it would be advantageous to include a dedicated section addressing non-operative treatments.

Here is a list of possible articles to draw inspiration from:

1)      Cameron KL, Driban JB, Svoboda SJ. Osteoarthritis and the Tactical Athlete: A Systematic Review. J Athl Train. 2016 Nov;51(11):952-961. doi: 10.4085/1062-6050-51.5.03. Epub 2016 Apr 26. PMID: 27115044; PMCID: PMC5224737.

2)      Rasteiro A, Santos V, Massuça LM. Physical Training Programs for Tactical Populations: Brief Systematic Review. Healthcare (Basel). 2023 Mar 28;11(7):967. doi: 10.3390/healthcare11070967. Erratum in: Healthcare (Basel). 2023 Sep 05;11(18): PMID: 37046894; PMCID: PMC10094380.

3)      Filardo G, Kon E, Longo UG, Madry H, Marchettini P, Marmotti A, Van Assche D, Zanon G, Peretti GM. Non-surgical treatments for the management of early osteoarthritis. Knee Surg Sports Traumatol Arthrosc. 2016 Jun;24(6):1775-85. doi: 10.1007/s00167-016-4089-y. Epub 2016 Apr 4. PMID: 27043347.

Author Response

Reviewer 2:

The work proposed by the authors is a comprehensive narrative synthesis of the concept of "osteochondral unit," with particular reference to the knee, its pathophysiology, and current treatment indications in case of damage to this unit. While theoretically focused on the so-called "tactical athletes," a closer examination of the manuscript reveals a lack of specific details essential for honing the discussed themes toward this distinct population. In other words, the concepts presented remain broad and lack the necessary refinement tailored explicitly to this category of athletes.

Moreover, the study is entirely oriented towards surgical treatment, overlooking the current understanding that knee osteochondral damage, particularly in case of chronic or overuse injuries, is often more effectively managed through non-surgical approaches including physical therapies, rehabilitation, pharmacological interventions, and infiltrative treatments (i.e.: hyaluronic acid, PRP, mesenchymal stem cell, etc.).

I suggest a meticulous revision of the manuscript, with a sharper focus on specific physical carachteristics and performance-related issues specific to tactical athletes. Furthermore, it would be advantageous to include a dedicated section addressing non-operative treatments.

Here is a list of possible articles to draw inspiration from:

1)      Cameron KL, Driban JB, Svoboda SJ. Osteoarthritis and the Tactical Athlete: A Systematic Review. J Athl Train. 2016 Nov;51(11):952-961. doi: 10.4085/1062-6050-51.5.03. Epub 2016 Apr 26. PMID: 27115044; PMCID: PMC5224737.

2)      Rasteiro A, Santos V, Massuça LM. Physical Training Programs for Tactical Populations: Brief Systematic Review. Healthcare (Basel). 2023 Mar 28;11(7):967. doi: 10.3390/healthcare11070967. Erratum in: Healthcare (Basel). 2023 Sep 05;11(18): PMID: 37046894; PMCID: PMC10094380.

3)      Filardo G, Kon E, Longo UG, Madry H, Marchettini P, Marmotti A, Van Assche D, Zanon G, Peretti GM. Non-surgical treatments for the management of early osteoarthritis. Knee Surg Sports Traumatol Arthrosc. 2016 Jun;24(6):1775-85. doi: 10.1007/s00167-016-4089-y. Epub 2016 Apr 4. PMID: 27043347.

The authors have added substantial detail to the article. With regard to the tactical athlete population, evidence of this specific population’s outcomes for each treatment modality is limited by the currently available literature. However, where appropriate, we have extrapolated clinical outcomes from other athletic populations when tactical athlete data is not available. Throughout the manuscript we have also provided several pertinent examples of important treatment considerations in this population of patients. We thank the reviewer for the three proposed citations, we have read through them and added in details or ideas from each as appropriate. This includes a section on non-operative treatment and injections.

Reviewer 3 Report

Comments and Suggestions for Authors

Dear authors,

In this review, you sought to present your approach for the knee joint preservation techniques in tactical athletes. Overall, the paper reads well and summarizes the most recent evidence on this topic. I have some minor comments that I believe they will improve the quality of your presentation.

Comment on Patellofemoral Joint Cartilage: I would suggest you mention what the average thickness of the patellar cartilage is here.

Figure 1. Could you please confirm that you have obtained permission to utilise the photo which appears in Figure 1? If not, then please explain.

Also, I would suggest you add some information regarding the changes the occur in osteoarthritis (eg increased water content) so the readers are aware of the differences.

Author Response

Reviewer 3:

In this review, you sought to present your approach for the knee joint preservation techniques in tactical athletes. Overall, the paper reads well and summarizes the most recent evidence on this topic. I have some minor comments that I believe they will improve the quality of your presentation.

Comment on Patellofemoral Joint Cartilage: I would suggest you mention what the average thickness of the patellar cartilage is here.

4-6 mm in males, added to the manuscript

Figure 1. Could you please confirm that you have obtained permission to utilise the photo which appears in Figure 1? If not, then please explain.

This is a figure that the first author designed with the help of our medical illustrator, no permission required.

Also, I would suggest you add some information regarding the changes the occur in osteoarthritis (eg increased water content) so the readers are aware of the differences.

The author’s thank you for this recommendation and have added these details to a new section on aging and degeneration.